# Impact of Long Covid on the school experiences of children and young people: a qualitative study

Alice MacLean [1], Cervantee Wild [2], Kate Hunt [1], Sarah Nettleton [3], Zoë C Skea [4], Sue Ziebland [2]

¹Institute for Social Marketing and Health, University of Stirling, Stirling, UK
²Nuffield Department of Primary Care Health Sciences, University of Oxford, Oxford, UK
³Department of Sociology, University of York, York, UK
⁴Health Services Research Unit, University of Aberdeen, Aberdeen, UK

**Correspondence to**
Dr Alice MacLean;
alice.maclean@stir.ac.uk

## ABSTRACT

**Objectives** To explore the impact of Long Covid (LC) on the school experiences of children and young people (CYP).
**Design** Qualitative study using narrative interviews.
**Participants** 22 CYP (aged 10–18 years, 15 female) with LC and 15 parents/caregivers (13 female) of CYP (aged 5–18 years) with LC.
**Setting** Interviews were conducted between October 2021 and July 2022 via online video call or telephone. Recruitment routes included social media, LC support groups, clinicians, community groups and snowballing.
**Results** Three key findings were identified. Finding 1: Going to school is a valued part of CYP's lives and participants viewed educational attainment as important for their future trajectories. Returning to school full time was highlighted as a key part of regaining 'normal life'. Finding 2: Attending school (in-person or online) with LC is extremely difficult; even a gradual return required CYP to balance the impact of being at and engaging with school, with the need to manage symptoms to prevent relapse. Often this meant prioritising school and rest over other aspects of their lives. Finding 3: School responses to CYP with LC were reported to be mixed and hampered by difficulties communicating with healthcare professionals during the pandemic and a lack of awareness of LC among healthcare and education professionals. Participants viewed supportive school responses as staff believing, understanding and taking them seriously, alongside schools offering tailored and flexible adaptations which allowed engagement with school while limiting any deterioration of symptoms.
**Conclusions** This study describes how LC affects the school experiences of CYP and generates recommendations for supportive school responses alongside supportive healthcare professionals. Further research could explore the approaches that facilitate a successful return to school for CYP with LC and investigate education professionals' perspectives on support they require to positively engage with returning pupils.

## INTRODUCTION

Children and young people (CYP) infected with SARS-CoV-2 are at lower risk of hospitalisation and mortality compared with adults[1] but, like adults, can experience symptoms which persist long after initial infection.[2–4] These ongoing symptoms are variously

### STRENGTHS AND LIMITATIONS OF THIS STUDY

⇒ To our knowledge, this is the first qualitative study to explore the impact of Long Covid (LC) on children and young people (CYP)'s experiences of school.
⇒ This study uses rigorous methods and draws on rich accounts from a diverse sample of CYP with LC, and parents of CYP with LC, across the UK.
⇒ A limitation of the study is that it does not include the perspectives of education professionals on their experiences of engaging with CYP with LC and the support they need to do this.
⇒ The composition of our sample prevented a systematic comparison of accounts by age, gender, ethnicity or social class, thus preventing investigation of whether CYP with LC experience similar expectations around responding to illness with stoicism, control and independence (especially while at school) which previous research has highlighted.

described as post-COVID-19 syndrome, post-COVID-19 condition and postacute sequelae of SARS-CoV-2 or Long Covid (LC), the name first used by adults with persisting symptoms.[5] There is an increasing research focus on LC in adults[6–9], but limited high-quality evidence on CYP and ongoing uncertainties, including the prevalence, risk factors, mechanisms and effective management of LC in CYP.[10–13] Here, we describe accounts from CYP and parents of the impacts of LC on education and school-related challenges and recommend ways that schools could support CYP with LC.

LC has been described as a 'frequent legacy of acute SARS-CoV-2 infection, affecting over 10% of patients' (p.1891)[7] with wide-ranging symptoms. Its clinical case definition in adults describes: 'a history of probable or confirmed SARS-CoV-2 infection, usually 3 months from the onset of Covid-19 with symptoms…that last for at least 2 months and cannot be explained by an alternative diagnosis'[14] and research suggests CYP have comparable symptom experiences to adults.[15] Children and adults report: many persistent symptoms affecting multiple body

systems,[11] with fatigue, headache, cognitive impairment and shortness of breath being common[2 16–18]; and symptoms that fluctuate in severity,[11] with new symptoms emerging potentially months after initial infection. A higher prevalence of persistent symptoms has been reported in females.[13 19] Accurately measuring symptoms of LC over time is challenging given the emergence of new variants of COVID-19, the potential impact of vaccines and the difficulty of ruling out the influence of other viral infections.[3 20]

Research into the social and academic impacts of LC on CYP is lacking and existing evidence is mixed. A Spanish study followed (for 5 months) 50 CYP with LC (defined as ongoing symptoms for 12+ weeks) and reported that 18% were unable to attend school, 34% had a reduced school schedule, 66% showed decreased school performance and 68% had stopped extracurricular activities.[15] Systematic reviews have highlighted heterogeneous findings relating to the impact of LC on CYP's lives. Pellegrino et al[12] included five studies which reported limitation in daily function affecting school attendance (in 10.5%–58.9% of participants across the studies). Franco et al[21] reviewed 25 studies which included well-being and recovery outcomes in CYP, five of which reported outcomes relating to 'changes in work/occupation and study (school attendance)'. The authors suggested 'most children with persistent symptoms reported no substantial impairment in their school functioning at 3–6 months follow-ups' (p.12). However, it has been noted that the evidence in CYP is limited, heterogeneous and largely based on low-quality studies.[12]

We found no qualitative studies on the impacts of LC on CYP's school experiences. However, research with CYP with chronic fatigue syndrome (CFS)/myalgic encephalomyelitis (ME), a condition with some similarities to LC,[22 23] highlights detrimental impacts on social, emotional and academic aspects of schooling. For CYP with CFS/ME, disrupted schooling 'has a significant impact on the self (resulting) in a shift from a perceived normal trajectory of academic achievement and independence to one that is uncertain' (p.10),[24] and some educational support and adaptations underpinned by evidence-based awareness-raising in schools have been recommended.[25–28]

## METHODS

This paper draws on narrative interviews undertaken to improve understanding of the experience of LC in households with CYP from diverse backgrounds. In this analysis, we draw on interviews with CYP (aged 10–18 years) with LC and parents/caregivers of CYP (aged 5–18 years) with LC, some of whom had LC themselves.

### Patient and public involvement

An advisory panel, including patient and public involvement representatives with lived experience of LC or of caring for a child or young person with LC, had input into all aspects of the study conduct, including content of the interview topic guide and recruitment methods.

### Recruitment and sampling

Recruitment routes included social media, LC support groups, clinicians, community groups and snowballing. We aimed for maximum variation sampling,[29] to capture diversity by age, gender, ethnicity, geographical location and social class. Potential participants were eligible if they, or the person they cared for, had self-identified ongoing symptoms 12+ weeks after initial COVID-19 infection. CYP were eligible if they were 10 years or over. Parents were eligible if their child with LC was 5 years or over. Age-appropriate information sheets were provided to potential participants.

### Data collection

Narrative interviews (n=37) were conducted (between October 2021 and July 2022) via online video call or telephone, and were video and/or audio recorded, depending on participant preference. Verbal consent was recorded at the start of their interview. Those under 16 years gave assent and their parent/caregiver gave proxy consent. Participants were offered a £30 voucher to thank them for sharing their time and experiences.

Interviews began with an open narrative where participants were asked to recount events since they/their child first experienced signs of COVID-19. The second part used topic guides (different for CYP and parent interviews) with prompts, including questions about how LC had affected school (see online supplemental file 1). Interviews typically lasted between 25 and 90 min, with some conducted over multiple shorter sessions to accommodate participant fatigue or other symptoms.

### Data analysis

Interviews were transcribed verbatim, checked for accuracy and imported into NVivo (March 2020 version) to aid organisation and coding of data. We used thematic analysis[30] to inductively code the data. After initial familiarisation with transcripts, we developed a coding framework of broad themes, which was refined throughout the coding process. Three researchers coded the transcripts (CW, ZCS and SN). All data coded to the broad 'school/education' theme were then further analysed (by AM) using the mind-mapping 'one sheet of paper' technique.[31] This process generated three subthemes, as described below with interview extracts (IE) to illustrate the range of views (longer extracts are presented in boxes 1–3). All names are pseudonyms. When quoting a parent who spoke in their child's interview, we use the convention 'mother of Gemma (16 years old, LC 19–24 months)'.

## RESULTS
### Participants

Table 1 displays the characteristics of the participants included in this analysis. Of the 15 parents of a child/young person with LC interviewed, 5 also had LC themselves.

**Box 1  '*I just want to be normal again*': education as a normal and valued aspect of life for CYP—illustrative interview extracts**

IE 1. '*I just want to go back to school. Right, I know a lot of kids can't say that, but [I want to] just go back to my normal routine*' (Holly, 14 years old, LC 0–6 months)

IE 2. '*I have really bad meltdowns where I just want to be back to normal(…)I do half days at school(…)go in at like 11am, and I come home and I just, I'm crying [and] 'I just want to be normal again'*' (Mae, 11 years old, LC 7–12 months)

IE 3. '*(S)he will always try…so she will get up in the morning, she will get herself ready(…)I'm like, 'You are not well,' and she's like, 'But I will try, Mummy.'(…)So she's very resilient, she doesn't ever let anything kind of hold her back(…)if she thinks she can try, she will*' (Evelynn, parent of 8-year-old, LC 13–18 months)

IE 4. '*[The hardest part is] not being able to go to school or like see people my age, socialise and everything. It's all like online for me now over like social media or messages(…)seeing other people(…)my age that are going out in school or doing all their exams [and] doing lots of things throughout the summer that I would like to be able to do, but I just can't. I think is that's quite hard.*' (Gemma, 16 years old, LC 19–24 months)
CYP, children and young people.

**Box 2  '*School is still a lot of energy*': experiences of returning to school with LC—illustrative interview extracts**

IE 1. '*[My daughter] would struggle, do a day [at school] and then she'd be in bed for two days. She'd crash, she couldn't get out of bed. She had pins and needles in her leg every time she stood up, she felt dizzy, nausea(…)so [it was like being] on a roller coaster*' (Olivia, parent of 11-year-old, LC 0–6 months)

IE 2. '*I just feel like ever since I've got [Covid], I've just lost a lot of my drive [for school] just because I'm always in such pain(…)constantly taking breaks and being in pain and you can't really concentrate if you were just, like, burning inside*' (Holly, 14 years old, LC 0–6 months)

IE 3. '*The last three days, my fingers have been really sore and it hurts to bend them and move them. So, [classroom assistant] writes for me*' (Josie, 10 years old, LC 0–6 months)

IE 4. '*Something's just…blocking my thoughts(…)I can't take in English, I can't analyse anything, it's like your sort of chain of thoughts is just completely, you know, broken, like I can't have more than one thought that leads to another one, it's just like my brain is just a cement block and it's just all messed up(…)it's so bad not being able to fully focus in class and struggling with the work and then not having the energy to do the homework(…)I've never really struggled in school before, I've always enjoyed learning, but(…)it really stopped all that [and] just feeling really like stupid(…)I couldn't do the work even if it was work that I could do before, [so] then in that way it impacts(…) your self-esteem and things like that*' (Erin, 15 years old, LC 0–6 months)

IE 5. '*[For my son] online learning at home wasn't an option either because he has quite a lot of brain fog so has a lot of difficulty with concentration and was finding learning new things seems to be really difficult, short-term memory, all sorts of things like that were really hard*' (Marissa, parent of 16-year-old, LC 7–12 months)

IE 6. '*I have a timeout pass [to excuse myself from class] and sometimes I have to go to the toilet [because] I get stressed because everyone else [in class] is working [and] you're like, 'What? I don't understand. What is happening?' and I feel like I get on people's nerves [when] I'm like, 'What?' and [they say] 'Well, you, you probably missed that,'*' (Mae, 11 years old, LC 7–12 months)

IE 7. '*[My friends] haven't really reacted, mainly they just like ask why I wasn't in school, and formed other friendships*' (Shay, 12 years old, LC 19–24 months)

IE 8. '*Some subjects I'm behind, but I'm still doing maths, English, science, computer science, I'm still making sure I'm on top of them really(…)But I've just left [other] subjects [because] it will take more brain power [and] be unneeded stress*' (Fred, 14 years old, LC 13–18 months)
LC, Long Covid.

In interviews with CYP, some parents chose to be present, either sitting beside their child or elsewhere in the room while the interview was conducted. When parents were present, some spoke very little in the interview and others contributed significantly. The sample included four families where more than one member took part individually, and one family contributed three separate interviews (two with children with LC and one with their parent).

Analysis identified three key themes: CYP's desire to get back to school; their experiences of being in school while still affected by LC; and schools' responses to their illness.

### '*I just want to be normal again*' (Mae, 11 years old, LC 7–12 months): education as a normal and valued aspect of life for CYP

Most CYP's accounts indicated that they had been too ill to attend school regularly or undertake online learning and some were not attending school at all when interviewed. GP and hospital appointments also caused frequent school absences. Analysis revealed a strong desire for their lives to return to the way they were before having Covid and CYP spoke about being able to go back to school as a major means of regaining some normality (see box 1-IE 1 and 2). Children's eagerness to return to school was also evident in parents' accounts (see box 1-IE 3).

Expressing their desire to return to school contradicts common stereotyping of children, and particularly teenagers, as lazy or reluctant to engage with school. Faye (14 years old, LC 13–18 months) said '*I would give so much to go back to school full-time. I miss it a lot*'. CYP's accounts portrayed school absence as making them stand out from their peers, going against the 'normality' of full-time school for people their age. Frequent or extended absences were described as stressful and isolating, leading to feelings of being left behind academically and socially (see box 1-IE 4). The unpredictable and variable nature of symptoms was particularly distressing because CYP did not know how long disruption to their schooling would last. Fred (14 years old, LC 13–18 months) said '*It's never ending […] you're in a maze and you turn around the corner and 'Oh, this is going to be the end,' but it's a dead end, that's what it feels like*'.

### '*School is still a lot of energy*' (Alana, 13 years old, LC 7–12 months): experiences of returning to school with LC

For many CYP we talked to, going back to school had not represented the hoped for 'return to normal' and they commonly highlighted extreme fatigue.

*after school, I would come home [and] sleep for ages [and] I've never done that before* (Faye, 14 years old, LC 13-18 months).

ð

**Box 3** **'*It's a mixed bag*':** Schools' responses to CYP with LC—illustrative interview extracts

IE 1. '…applying a lot of pressure for us to send her back in and we were saying 'Yeah, we want her to be in school but there's a huge outbreak [of Covid] and… and you're offering her no protection.' The attitude of the headteacher was, 'I'm not going to talk to you, I've answered your questions, I'm following government guidance, I'm not going to talk to you anymore about this,' so she completely closed down the conversation around health and safety' (Laura, parent of 6-year-old, LC 7–12 months)

IE 2. '[All the parents on Long Covid support groups] have the same problem. If the doctor doesn't provide any letters(…)it's not that the school doesn't want to support a child, but they need to have some kind of evidence [and] if the paediatrician doesn't recognise that you know, yes the child has a reason for a… tired for a reason they would say, 'Well, I don't see anything why there are… you know all the blood test results come back fine, there is no other reason,' then the school has no backup to say 'Yes, OK we can authorise this.'(…)[Once] I'd managed to get a letter from a paediatrician then [the school] said, 'Yes, you know, how many hours can he do? What can he do?'' (Freja, parent of 12-year-old, LC 7–12 months, interviewed in Spring 2022)

IE 3. 'I've got [an Artificial Intelligence-based] robot now which can go into school [when I'm off](…)I do like a day on like a Tuesday and a Thursday on the robot. So, if I'm tired, I can be in my bed, resting and just listening to [what is being taught in class through the robot]. So, I'm still learning stuff' (Faye, 14 years old, LC 13–18 months)

IE 4. '… [because my daughter] was on the first floor [the] school had to move around six classes to be able to get [her wheelchair] into a ground floor classroom(…)the school couldn't have done more' (Jan, parent of 10-year old, LC 0–6 months)

IE 5. '[My teachers are] trying to get me back into school(…)full-time again, but like it's hard for me(…)some of the teachers aren't [supportive] because they're telling my mum they're going to fine her(…)if I don't go back full-time(…)she'd got a letter from the GP saying that there was something wrong with me and I've been sick [but] I don't think they understand the effect [Long Covid] has on you(…)I think they think that I'm not going to school because I don't want to [but] I liked going to school, like I never had a problem with it.' (Hala, 15 years old, LC 7–12 months, interviewed in Summer 2022)

CYP, children and young people; LC, Long Covid.

*I couldn't really do anything [with friends] at break. I was just resting. I struggled going up the stairs. I can't do PE. Yeah, I just feel tired after every lesson* (Rory, 13 years old, LC 7-12 months).

Attempts to return to school invariably led to 'crashes', 'huge relapses' or feeling 'exhausted', followed by needing more time off to recover (see box 2-IE 1).

Navigating school buildings was difficult and participants said it made symptoms worse.

*school is still a lot of energy because I go to a very big school […] so that's a lot of walking about* (Alana, 13 years old, LC 7-12 months).

*I've got third floor, second floor, first floor, the ground floor, I have to go up and down stuff [and] I get tired* (Rohaan, 12 years old, LC 0-6 months).

**Table 1** Participant characteristics (n=37)

| Age (years) | | Sex | |
|---|---|---|---|
| 10–12 | 8 | Male | 9 |
| 13–18 | 14 | Female | 28 |
| 19–30 | 0 | | |
| 31–40 | 6 | | |
| 41–50 | 7 | | |
| 51+ | 1 | | |
| Missing | 1 | | |

| Ethnicity | | Time child affected by LC symptoms (months) | |
|---|---|---|---|
| White British | 21 | 0–6 | 12 |
| White other | 3 | 7–12 | 15 |
| British Asian | 8 | 13–18 | 4 |
| Black Other | 1 | 19–24 | 5 |
| Mixed race | 1 | 25+ | 1 |
| Other | 3 | | |

LC, Long Covid.

*[My classroom] used to be upstairs and that was the hardest bit because before I stopped walking [and got a wheelchair], I had to go upstairs and my knees were so sore* (Josie, 10 years old, LC 0-6 months)).

As well as fatigue, other physical symptoms impacted participants' ability to learn (see box 2-IE 2 and 3).

Many highlighted difficulties caused by cognitive impairment, such as Shay (12 years old, LC 19–24 months) who said '*I can't concentrate with reading*' and Fred (14 years old, LC 13–18 months) who stated '*I sometimes get brain fog, I'm sometimes just lost*'. Those who previously enjoyed school described these difficulties as particularly distressing (see box 2-IE 4). For many, learning online from home was not an appropriate solution. Gemma (16 years old, LC 19–24 months) who was no longer going to school when interviewed, said '*[for two months] I was trying to do a full week at school from home while not feeling well and I just couldn't cope with it, so I was removed from all my classes*'. Those with brain fog and fatigue struggled to follow online learning (see box 2-IE 5). There were also social and emotional difficulties involved in returning to school. Some found it distressing and isolating to feel they were falling behind their peers academically, and some described how their absences from school contributed to the disintegration of friendships (see box 2-IE 6 and 7).

CYP who were able to attend school, at least part time, struggled to make a successful return while managing their symptoms. Secondary school pupils spoke about being aware of the importance of education for their future and wanting to do well in exams to keep further education and career options open ('*I want to get good GCSEs*' (Layla, 14 years old, LC 7–13 months)). Balancing the impact of working hard at school with managing their

symptoms and preventing relapses was difficult. Sacrifices and prioritisation were often recounted, such as concentrating on fewer school subjects (see box 2-IE 8) or restricting social activities to conserve energy for going to school. Erin (15 years old, LC 0–6 months) said '*I'm not doing anything outside of school really […] I go to school, I go to sleep, […] which is difficult*'.

### '*It's a mixed bag*' (Fred, 14 years old, LC 13–18 months): schools' responses to CYP with LC

Parents spoke about alerting their child's school to the health challenges they were experiencing and explaining their absences. This could be difficult due to the varied and unpredictable nature of symptoms, as indicated by Freja (parent of 12-year-old, LC 7–12 months): '*I said to [son's school], 'No, he doesn't have fever, he is just exhausted, I can't send him to school, I'm sorry,' so of course [the school is] concerned*'. One parent, whose child caught Covid in Winter 2021, said she had tried to explain LC to her daughter's school '*because it's new to them as well*' (Angela, parent of 8-year-old, LC for 0–6 months). Parents understood schools require explanations for absences, ideally 'backed-up' by validation from healthcare professionals. However, the pandemic had made it harder to facilitate communication between healthcare professionals and schools.

When Covid cases were high in school, some parents preferred their child to stay away to avoid reinfection:

*cases are so high [that] I've got no interest in sending [my children to school]* (Izzy, parent of 12-year-old, LC 19-24 months).

*[son] went back [to school] again [and] then we had to miss the last couple of days [before Christmas holidays] because cases were going through the roof* (Ross, parent of 13-year-old, LC 19-24 months)).

A paediatrician's reported advice that school was '*absolutely the best place*' for her daughter, was unwelcome to a parent who felt her concerns about reinfection and its impacts were dismissed:

*I kept saying, 'But what happens if she goes back to school and gets Covid [again] on top of how she's feeling?* (mother of Gemma, 16 years old, LC 19-24 months).

Another mother, whose daughter caught Covid in Spring 2021, said school had ignored her fears about the risk of reinfection and she felt they were applying pressure to send her daughter back to school (see box 3-IE 1).

A perceived lack of integration of care across health and education settings was compounded by the fact that LC in CYP was a new condition that was neither widely recognised nor well understood. Parents suggested that the absence of a formal diagnosis meant schools and education professionals were limited in the support and adaptations they could offer (see box 3-IE2). Even when contact was made between health and education practitioners, it was then hard for parents and CYP to plan how much and how often they could attend school because

symptoms were unpredictable, varying day-to-day or week-to-week and activity needed to be balanced against potential relapses.

Other participants recounted positive experiences once a link had been made between school and a healthcare professional who suggested ways to manage symptoms at school (eg, attending part-time, not doing physical education, frequent rest breaks). Some schools were described as responsive, assertive, supportive and flexible, by putting various adjustments in place (see box 3-IE 3 and 4). Parents and CYP also mentioned understanding and supportive teachers:

*My art teacher has been really good [and] worked out what work absolutely needed to be done [and] came up with the idea of basing my whole art project around Long Covid* (Molly, 16 years old, LC 0-6 months).

*[The teacher's] wife had chronic fatigue syndrome so he's really understanding and he's amazing with [my daughter]* (Olivia, parent of 11-year old, LC 0-6 months).

CYP and parents wanted to feel that school staff believed them and took the impact of LC seriously. This was the case for Evelynn (parent of 8-year-old child, LC 13–18 months) who said '*[her teachers] are really flexible if she needed a break, she can go and sit down: there's no questions asked*'.

However, a few parents and CYP said that schools had not implemented systems to facilitate reintegration.

*I had to tell [my teachers about Long Covid] myself [because] all school told them was that I was going part-time* (Faye, 14 years old, LC 17-24 months).

*[During PE] they forgot I couldn't do stuff [and] I just had to sort of watch them do fun stuff* (Rory, 13 years old, LC 7-13 months).

Others, including some interviewed in 2022, described a lack of awareness and understanding of LC among school staff. Fred (14 years old, LC 13–18 months), interviewed in Spring 2022, described his teachers' responses as a '*mixed bag*'; some understanding teachers said, '*don't push yourself*' while others were '*the opposite and want me to do the same things and the same tests [as other pupils]*'. Even when healthcare professionals had written to the school, there were still accounts of disbelief, a lack of understanding and even threats of fines for non-attendance (see box 3-IE 5).

## DISCUSSION

Our findings highlight the importance of school to CYP with LC and how returning to school was central to CYP's much hoped for 'return to normal'. CYP described absences from school as stressful and isolating and placed high importance on returning to school full-time. However, re-engaging with school could lead to relapses, and further absences, meaning CYP had to learn how to deal with the demands of school without pushing themselves to 'crashing' point. Often this meant prioritising

> **Box 4 Recommendations for schools and healthcare professionals supporting CYP with Long Covid (LC)**
>
> 1. Show that you believe, understand and are willing to help CYP with LC attempting to return to school and validate their strong desire to return to school and a 'normal life'
> 2. Recognise the difficulties parents of CYP with LC may face in facilitating school contact with healthcare professionals in the face of pressures on healthcare services, and in future-planning for attendance due to the unpredictable nature of LC
> 3. Raise awareness and knowledge of LC among the school community (staff and CYP)
> 4. Communicate regularly with CYP with LC about how they are coping with school alongside their illness
> 5. Offer a range of adaptations which can be tailored to the individual and changing needs of CYP with LC. For example:
>     a. Reduced timetables and/or prioritisation of fewer subjects.
>     b. Rest/'time-out' passes.
>     c. Allow the use of lifts and other means of alleviating the physical impacts of moving around school buildings.
>     d. Consider the use of appropriate technology (eg, artificial intelligence-based robots) to facilitate engagement with school.
> 6. Ensure school staff are advised of CYP who have LC and are aware of the adaptations available to them.
> 7. Consult school staff on their experiences of supporting CYP with LC and what they need to facilitate positive engagement with this group of pupils.

school and rest over all other aspects of life. The various school responses to LC that participants described have highlighted the importance of validation of CYP's experience of LC by healthcare professionals and informed recommendations for supporting CYP with LC at school to minimise adverse educational, social and mental health sequelae of having LC in childhood and adolescence (see box 4). These recommendations for practice foreground the lessons arising from dealing with the ongoing symptoms of COVID-19 in CYP and highlight ways to respond to other long-term health conditions and the potential impacts of future pandemics on school pupils.

To our knowledge, this is the first qualitative study to explore the impact of LC on CYP's experiences of school. A further strength of our study is that it uses rigorous methods and draws on rich accounts from a diverse sample of CYP with LC across the UK. A limitation is that it does not include the perspectives of education professionals on their experiences of engaging with CYP with LC and the support they need to do this. The composition of our sample prevented a systematic comparison of accounts by age, gender, ethnicity or social class, thus preventing investigation of whether CYP with LC experience similar expectations around responding to illness with stoicism, control and independence (especially while at school) which previous research has highlighted.[32] It is also important to acknowledge that participants' accounts may have been impacted by the fact that they were being filmed/audiorecorded for a website. However, all participants were given the option of safeguarding their identity by making their data fully anonymous.

There are few existing studies with which to compare our findings, although there are some parallels with studies of the impact of CFS/ME and other long-term health conditions on school experiences, particularly the importance of school and its centrality to CYP's hopes to return to a 'normal' trajectory.[24 27 33 34] School absence because of LC marked CYP out from peers and they reported feeling stressed, isolated and worried about falling behind academically and socially. These negative emotions were compounded by not knowing how long this 'break from normality' and uncertain trajectory would continue, as has been reported for CYP with CFS/ME, juvenile idiopathic arthritis and other long-term conditions.[24 35 36] Echoing the 'ripple effect' of CFS/ME on CYP's social, emotional and academic functioning,[25] our findings demonstrate the impact of LC on CYP's participation in many aspects of school life, highlighting their distress about their inability to fully reintegrate with peers in the way they had hoped. As CYP placed a high value on education and often prioritised school (and rest) over all other aspects of their lives, there is a need to find ways of helping CYP with LC and other long-term conditions to minimise school demands so that they can also participate in other aspects of their lives.[28] As reported for CYP with CFS/ME,[27 28] we found that schools' responses were experienced as particularly supportive when participants felt believed and understood by school staff (often following validation of the CYP's condition by a healthcare professional) and when schools instigated tailored and flexible adaptations to help CYP engage with school while preventing relapse. The recommendations for healthcare and education professionals stemming from our findings are underpinned by values such as empathy, respect, openness and flexibility. They echo the 'approaches of positive schools' and key actions in the role of schools and teachers in engaging young people with health conditions outlined by Hopkins et al[37] (p.32).

A notable difference from research on other illnesses is that LC emerged as a new condition during a global pandemic. There is an expanding literature on the processes that adults with LC have gone through to recognise their symptoms as legitimate and worthy of investigation, support and treatment.[5 8 32] However, little is known about equivalent experiences of CYP with LC and their parents/caregivers. Indeed, it has been suggested that resistance to recognising LC as a clinical entity has been even more pronounced for CYP.[18] Research on paediatric CFS reports that CYP and parents still 'often report feeling misunderstood and disbelieved by medical and educational services' despite CFS being a defined diagnostic entity since 1991 (p.12).[38] Studies of CYP with CFS/ME highlight the importance of diagnosis and healthcare professional validation in legitimising the illness.[27 28] Our findings suggest that harnessing this 'power of diagnosis' has been especially problematic for CYP with LC due to the novelty of LC and recency of its emergence within a time of unprecedented pressures on healthcare services, which limited parents' ability to facilitate links between

healthcare and education professionals. While Similä *et al*[28] found that online teaching was perceived as helpful by CYP with CFS/ME during periods of lockdown, this was not always the case for CYP with LC in our study. Again, this highlights the need for further research on how educational adaptations can be tailored to the ways that LC symptoms impact on CYP's abilities to learn to minimise adverse health, social and educational outcomes in later life. For some parents in our study, a further tension existed around whether school was a safe place for CYP with LC, particularly when high case numbers in schools fuelled concerns that reinfection with COVID-19 might exacerbate or prolong LC symptoms. This novel aspect of our findings also requires further exploration. On the whole, it is important to learn from experiences of the COVID-19 pandemic in order to inform responses to future pandemics.

Our findings suggest ways that schools can support CYP with LC and highlight the need to raise awareness of LC among healthcare and education professionals. However, there is a need for further research on appropriate educational and social adaptations for CYP with LC, and increased knowledge and understanding of LC in schools and the most supportive ways that schools can respond. Further research might explore experiences by age, gender, social class and ethnicity to identify where interventions could be focused. Research with educational professionals could investigate how best to facilitate positive engagement with CYP with LC and their parents. Overall, it is important that CYP with LC are supported to engage with school in ways that facilitate recovery and minimise the impact of LC on their lives, physical and mental health and education longer term.

**Acknowledgements** We would like to thank all the children, young people and parents who took part in interviews, especially as many of our participants were still very affected by their Long Covid symptoms and had limited physical and cognitive resources. We would like to thank the funders of our research and our colleagues at the Health Experiences Research Group in Oxford. Thank you especially to Dr Anna Dowrick for helpful comments on an earlier draft.

**Contributors** SZ and KH developed the study, with input from CW, SN, ZCS, AM and the research advisory group which included CYP with LC and parents of CYP with LC. CW led the day-to-day management of the project, supported by SZ, KH, SN, AM and ZCS. The qualitative interviews were conducted by CW, SN, KH, AM and ZCS; all are highly experienced qualitative researchers. CW, ZCS and SN coded the interview transcripts. AM further analysed all data relating to school/education using the mind-mapping 'one sheet of paper' (OSOP) technique. AM drafted the manuscript, and all authors provided critical comments on drafts, and read and approved the final manuscript. All authors contributed to revisions of the paper. SZ as guarantor accepts full responsibility for the finished work and the conduct of the study, had access to the data, and controlled the decision to publish.

**Funding** This work is independent research funded by the National Institute for Health and Care Research (NIHR) (COV-LT2-0005).

**Disclaimer** The views expressed in this publication are those of the author(s) and not necessarily those of NIHR or The Department of Health and Social Care.

**Competing interests** None declared.

**Patient and public involvement** Patients and/or the public were involved in the design, or conduct, or reporting, or dissemination plans of this research.

**Patient consent for publication** Not applicable.

**Ethics approval** This study involves human participants and ethical approval was granted by Berkshire Ethics Committee (12/SC/0495). Participants gave informed consent to participate in the study before taking part.

**Provenance and peer review** Not commissioned; externally peer reviewed.

**Data availability statement** Data are available on reasonable request. The University of Oxford holds the copyright for the full interview transcripts and may grant data sharing permission on request.

**ORCID iDs**
Alice MacLean http://orcid.org/0000-0002-9650-2376
Cervantee Wild http://orcid.org/0000-0001-5377-6222
Kate Hunt http://orcid.org/0000-0002-5873-3632
Sarah Nettleton http://orcid.org/0000-0002-5184-2764
Zoë C Skea http://orcid.org/0000-0003-4685-4266
Sue Ziebland http://orcid.org/0000-0002-6496-4859

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
