## [Reviewer comments · BMJ Open]

ARTICLE DETAILS

TITLE (PROVISIONAL)	The impact of Long Covid on the school experiences of children and young people: a qualitative study.
AUTHORS	MacLean, Alice; Wild, Cervantee; Hunt, Kate; Nettleton, Sarah; Skea, Zoë; Ziebland, Sue

VERSION 1 – REVIEW

REVIEWER	Clemensen, Jane Syddansk Universitet Det Sundhedsvidenskabelige Fakultet, HC Andersen Children Hospital
REVIEW RETURNED	11-Jul-2023

GENERAL COMMENTS	Important paper we can learn from due to future pandemics. I therefore miss an "implication for practice" - some discussion of how we can use the results in common future situations.
--

REVIEWER	Chew-Graham, Carolyn University of Keele I am aware of this work and am involved in an SPCR-funded study into Long Covid in children and young people.
REVIEW RETURNED	14-Jul-2023

GENERAL COMMENTS	This is an excellent manuscript reporting narratives of children and young people, and parents/carers about covid, Long Covid and school. The authors highlight the involvement of their patient/public advisory panel. Description of analysis of qualitative data was adequate. I have one comment about style - the data extracts are buried in the body of the Findings section which makes it difficult for the reader to work through. Some of the data extracts are rather short. Could the authors clarify whether the text in inverted commas of each of the theme headings are data extracts? This needs to be made clear. In the Discussion, the authors include discussion of the need to 'harness the power of diagnosis' but this has not been referred to in the Findings section. I would suggest that new findings from the data analysis should not be introduced in the Discussion section of a manuscript.
---

VERSION 1 – AUTHOR RESPONSE

Reviewer 1

Important paper we can learn from due to future pandemics. I therefore miss an "implication for

practice" - some discussion of how we can use the results in common future situations.

In terms of highlighting the implications of our findings for practice, we have generated a list of recommendations for supportive school and healthcare professional responses to children and young people with Long Covid. These recommendations are outlined in Box 4 and highlighted in the first paragraph of the Discussion section. We have added a sentence to this paragraph to highlight that the lessons arising from the COVID-19 pandemic can inform our responses to future pandemics. The sentence reads: "These recommendations for practice foreground the lessons arising from dealing with the ongoing symptoms of COVID-19 in children and young people and highlight ways to respond to other long-term health conditions and the potential impacts of future pandemics on school pupils." We have reiterated the importance of learning from the COVID-19 pandemic in the penultimate paragraph of the Discussion by adding this closing sentence: "On the whole, it is important to learn from experiences of the COVID-19 pandemic in order to inform responses to future pandemics."

Reviewer 2

I have one comment about style - the data extracts are buried in the body of the Findings section which makes it difficult for the reader to work through. Some of the data extracts are rather short.

In writing our findings we have tried to present the range and diversity of perspectives and experiences in participants' accounts whilst not exceeding the journal word limit. This is why we chose to present multiple short quotes in brackets within the text of the findings and longer quotes in text boxes. However, we appreciate this reviewer's feedback on style and to address the sense that findings are buried, we have tried, where possible, to foreground quotes which were previously in brackets by presenting them separately and indenting them from the main body of the findings text.

Could the authors clarify whether the text in inverted commas of each of the theme headings are data extracts? This needs to be made clear.

We confirm that the text in inverted commas in each of the theme headings are data extracts. We have made this clear in the revised manuscript by italicizing the quotes and attributing them to participants.

In the Discussion, the authors include discussion of the need to 'harness the power of diagnosis' but this has not been referred to in the Findings section. I would suggest that new findings from the data analysis should not be introduced in the Discussion section of a manuscript.

We have added a sentence to the third section within the findings to make it clearer that there was evidence in the data of parents referring to the implications of the absence of a formal diagnosis of Long Covid. This sentence reads: "Parents suggested that the absence of a formal diagnosis meant schools and education professionals were limited in the support and adaptations they could offer (see Box 3- IE2)." Given the addition of this sentence, we have retained the sentence in the Discussion which reads: "Our findings suggest that harnessing this "power of diagnosis" has been especially problematic for CYP with LC due to the novelty of LC and recency of its emergence within a time of unprecedented pressures on healthcare services, which limited parents' ability to facilitate links between healthcare and education professionals."